# Analysis of the key influencing factors of China's cross-border e-commerce ecosystem based on the DEMATEL-ISM method

**Xiaodan Xi** [1,2]*, **Mingxia Wei**[2], **Brian Sheng-Xian Teo**[1]

**1** Post Graduate Centre, Management and Science University, Shah Alam, Malaysia, **2** School of Management, Henan University of Technology, Zhengzhou, China

* 1162845552@qq.com

**Data Availability Statement:** All relevant data are within the paper and its Supporting Information files.

**Funding:** The author(s) received no specific funding for this work.

## Abstract

Systematically analysing the relative importance and hierarchical relationships among the influencing factors of the cross-border e-commerce ecosystem holds rich theoretical value and practical significance for the development of this ecosystem. A total of 19 influencing factors covering four aspects affecting the cross-border e-commerce ecosystem are identified by means of the relevant literature, web pages, research, and discussions with relevant experts and scholars, and the decision-making trial and evaluation laboratory (DEMATEL) and interpretative structural modeling (ISM) method is used to analyse the cause-effect correlation of each factor and to obtain a cause-effect diagram and a multi-level recursive structure model. The results show that three factors, i.e., the e-commerce platform development level, cross-border e-commerce competitiveness, and the cross-border e-commerce transaction scale, have a greater degree of influence on the other influencing factors. Additionally, three factors, i.e., the information development level, GDP, and cross-border online shopping demand, are vulnerable to the influence of the other factors. The level of cross-border e-commerce platform development, cross-border e-commerce competitiveness, and inter-firm competition are the root factors and occupy an important position in the cross-border e-commerce ecosystem as influencing factors and influence the stability of the cross-border e-commerce ecosystem by affecting the other factors.

## 1. Introduction

Due to the COVID-19 pandemic, the current world economy is extremely complex, and the growth in China's traditional import and export trade is slightly sluggish. However, with the continuous development of information technology and the deepening of trade globalization [1,2], cross-border e-commerce, which is a new form of business and a new model, has shown unique advantages in driving industrial transformation and economic restructuring [3]. Under the incentive of continuous policy dividends and based on the enormous consumer group, China's cross-border e-commerce continues to develop [4]. With the increasing improvement in the Internet, communications, and logistics infrastructure, e-commerce in China is gradually showing a leading global trend [5]. According to the E-Commerce Research

**Competing interests:** The authors have declared that no competing interests exist.

Center, in the first half of 2022 alone, the size of the cross-border e-commerce market exceeded 7 trillion yuan, accounting for 35.85% of the total value of imports and exports of China's trade in goods of 19.8 trillion yuan. Around cross-border e-commerce, a series of emerging enterprises in logistics, finance, payment, and other fields has been formed, becoming a new growth point for the Chinese economy [6].

However, different from traditional international trade and domestic e-commerce, the development of cross-border e-commerce has its particularities and complexity. Cross-border e-commerce is conducted against the backdrop of international business activities, and participating in the main body mainly involves two or more different countries [7]. The complexity of trading in e-commerce is also increasing, including the cultural background and mode of payment [8]. The policy differences between the complexity of factors [9] such as any one of these links could affect the realization of cross-border transactions [10]. Therefore, how to effectively promote the orderly development of the cross-border e-commerce industry is an urgent problem that currently needs to be solved.

Many scholars have introduced ecosystem theory into the study of the e-commerce industry, providing a new research perspective [11]. In addition, combining the nature of the biological and environmental ecological system [12,13], the cross-border e-commerce ecosystem has an electronic business platform as the core and uses modern information technology, and cross-border e-commerce activities include logistics enterprises, foreign trade services, and cross-border payments. Such a series of cross-border trade enterprises and users gathered together on an electronic business platform play their respective roles and constantly communicate with their environment, thus forming a complex ecosystem with wide coverage and rich species [14,15]. In addition, with the continuous development of information technology, the cross-border e-commerce ecosystem can not only improve the overall efficiency of platforms but also further utilize the advantages of cross-border e-commerce [16], and its importance has received increasing attention.

## 2. Literature review

The concept of a cross-border e-commerce ecosystem first appeared in industry. Subsequently, scholars have gradually analysed cross-border e-commerce from the ecosystem perspective, which is a new theoretical perspective. Since research started relatively late, research on the cross-border e-commerce ecosystem has mainly focused on its constituent elements, connotation and characteristics, evolutionary path and other aspects.

According to several studies retrieved from the database, Zhang Xiaheng took Jingdong as a case to analyse and propose the components of the cross-border e-commerce ecosystem [3], laying the foundation for research by later scholars. Wu Min, based on the "Internet +" perspective, pointed out the problems existing in the process of constructing the cross-border e-commerce ecosystem, provided a framework for optimizing this ecosystem, and proposes optimization countermeasures based on the aspects of logistics and the market operation system [17]. He Juan [11] and her colleagues have grounded themselves in the ecosystem of cross-border e-commerce trade in the shipping industry. They have combined the theories of trade ecosystems and relevant economic theories and introduced a model of symbiotic co-evolution dynamics to explore the symbiotic co-evolution issues of cross-border e-commerce. Zhang Xiangxian et al., based on the new perspective of system theory and information ecology theory, analysed the meaning and characteristics of the e-commerce ecosystem and further constructed an e-commerce information ecosystem, which has important reference value for the study of the cross-border e-commerce ecosystem [18]. Cao Wujun et al. used a case study to analyse and build a cross-border e-commerce ecosystem dominated by logistics enterprises

and to reveal the general law of evolution of this ecosystem [19]. Research based on different enterprises leading the cross-border e-commerce ecosystem also includes the Xue Chaogai team [20]. Based on the ecosystem concept, they built a cross-border e-commerce ecosystem dominated by payment enterprises. Regarding the evolutionary path of the cross-border e-commerce ecosystem, Ji Shuxian and Li Junyan, based on an in-depth analysis of the Alibaba ecosystem case [21], argued that the evolution of the e-commerce ecosystem can be divided into four stages: the initial formation, development and growth, collaborative maturity and decline. Zhang Henan and Xu Zhengliang, based on the perspective of symbiosis theory [22], not only provided the important components of the cross-border e-commerce ecosystem but also provided the symbiotic evolutionary path of this ecosystem. With the deepening of research, scholars have not only discussed its evolutionary path but also have shown a strong interest in its evolutionary dynamics. Li Juanbo et al. used the ANP to construct a key dynamic model of the e-commerce ecosystem and summarized and analysed the dynamic mechanism and evolutionary path of the cross-border e-commerce ecosystem from the dimensions of politics, society and culture, the external economic environment, and the technological infrastructure environment [23]. Based on their research results, Xue Chaogai et al. further identified the key factors affecting the cross-border e-commerce ecosystem and pointed out that policy support, the technological development level and enterprise internal management are the key evolutionary dynamics [20]. However, based on a large number of studies, the concept of a cross-border e-commerce ecosystem was introduced late, and there are still many gaps in the relevant research.

Based on the above, previous research on the cross-border e-commerce ecosystem has established a preliminary foundation. However, further research has revealed the following shortcomings: 1. Lack of comprehensive summarization and analysis of the factors influencing the development of the cross-border e-commerce ecosystem; 2. There are still many research gaps in the interactive relationships among the relevant influencing factors; 3. There is a lack of quantitative analysis of the relevant influencing factors. Accordingly, on the basis of relevant research, this study intends to comprehensively identify the influencing factors affecting the development of the cross-border e-commerce ecosystem from the internal aspects of enterprises, policy support, the economic basis and the external environment through text mining and expert discussion, and it introduce the relevant theories of the DEMATEL-ISM methods. It analyses the interrelationship between many influencing factors and their degree of effect and successfully identifies the key factors, the causal relationships between factors and the hierarchical influence structure, as well as the dominant influence transmission path that has a great impact on the cross-border e-commerce ecosystem and is widely related. It does so to provide a reference and decision-making support for the operation and management practice of the China's cross-border e-commerce ecosystem and to enrich the relevant theories on the cross-border e-commerce ecosystem.

## 3. Introduction to the DEMATEL-ISM method

### 3.1 Method definition and analysis

The DEMATEL method, which is a method used to analyse the importance of the internal factors of complex systems, is mainly based on graph theory and matrix tools to analyse the elements of the system [24–26]. It helps to determine the nature of the research problem, which can be used as the basis for decision making. The DEMATEL method mainly aims to solve complex problems by directly comparing the influence relationship between factors in the problems [27]. Then, the direct and indirect causal relationship and influence intensity between factors are obtained through a matrix operation. Based on the calculation results, the

visual matrix and relationship diagram are used to express the relationship between the factors of complex problems and their influence degree [28,29].

At present, the ISM method is widely used in modern systems engineering research. It is a technical analysis method of structural modelling [30,31]. This method analyses the relationship between the elements of the system based on graphical theory and a matrix directed graph, constructs the hierarchical structure diagram of the abstract factors in the system, and identifies the surface factors and fundamental factors in the system [32]. It can transform obscure thoughts and ideas into intuitive structural models. It has a good effect on problems with many variables and complex relationships [33].

In order to promote and enhance the development of the cross-border e-commerce ecosystem, this study provides a comprehensive summary and analysis of the factors that affect its development, and further analyses the interactive relationship and degree of the influencing factors, identifies key factors, causal relationships between factors, and the hierarchical influence structure. However, traditional research methods have limitations, such as the PSR theory framework [34], data envelopment analysis method [35], analytic hierarchy process [36], and ISM method [32], have limitations. The PSR theory framework method is a qualitative method with poor objectivity. The data envelopment analysis method requires a high sample size. The use of the ISM method alone can only identify the hierarchical relationship between factors. Therefore, this study adopts a combination of the DEMATEL and ISM methods to achieve complementary advantages, clarifying the importance and causality of each factor in the system, and deepening the understanding of the logical relationships and hierarchical structures between factors.

Specifically, this study combines the comprehensive influence matrix in DEMATEL with the unit matrix to obtain the overall influence matrix, and transforms it into the reachable matrix required by ISM through calculation [37]. Compared with using ISM alone, this method not only shows the relationship between influencing factors, but also reflects the strength of interaction between them. The DEMATEL method is micro-oriented while the ISM method is macro-oriented [38]. Integrating the DEMATEL and ISM methods for research can complement advantages, improve computing efficiency, and comprehensively analyze the influencing factors of the cross-border e-commerce ecosystem from the levels, paths, and degrees of influence. This combination method avoids the shortcomings of DEMATEL in expressing the interrelationships and logical relationships between influencing factors and the shortcomings of ISM in accurately analyzing the degree of influence of each influencing factor on the complex system.

## 3.2 DEMATEL-ISM method of model Building (implementation steps)

First, the DEMATEL method is used to identify the core elements of the system and the degree of mutual influence by constructing a causal relationship diagram. On this basis, the ISM method is used to identify the surface factors and fundamental factors in the system. The combination of the two methods can not only comprehensively clarify the logical relationship and hierarchy of the system but also identify the core elements of the system. The process of analysis of the influencing factors of the cross-border e-commerce ecosystem based on the DEMATEL-ISM method is shown in Fig 1.

The cross-border e-commerce ecosystem research group established the index system through an investigation of the relevant literature, books, web pages and materials, as well as discussion with relevant experts and scholars. The implementation steps based on the DEMATEL-ISM model are as follows:

1. Determine the influencing factors of the cross-border e-commerce ecosystem, denoted as $S_1, S_2, S_3, \ldots, S_{23}$.

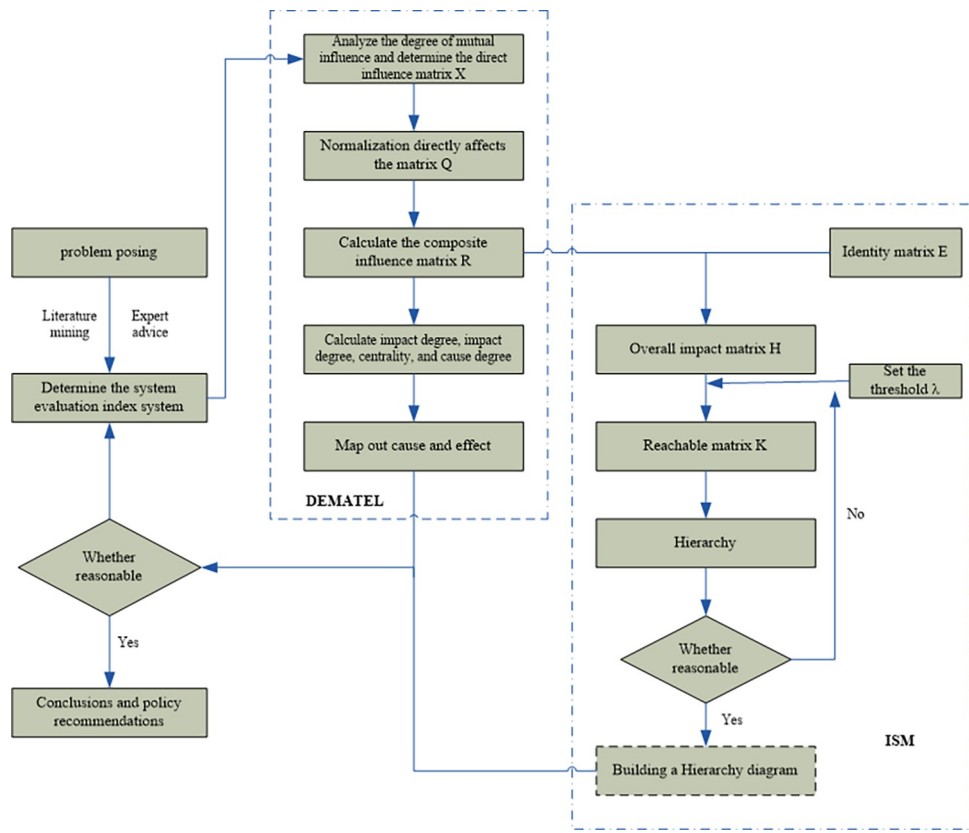

**Fig 1. Process of analysis of the influencing factors of the cross-border e-commerce ecosystem based on the DEMATEL-ISM method.**

2. Describe the degree of influence of factor $i$ on factor $j$ and express it as $x_{ij}$. The details are as follows:

$$x_{ij} \begin{cases} 0, \text{no influence} \\ 1, \text{little influence} \\ 2, \text{general influence} \\ 3, \text{strong influence} \\ 4, \text{very strong influence} \end{cases}, i = 1, 2, \cdots 19; j = 1, 2, \cdots 19.$$

3. Construct the direct influence matrix. With the degree of influence between factors as the element, direct influence matrix $X$ is constructed as follows:

$$X = \begin{bmatrix} x_{1,1} & x_{1,2} & \cdots & x_{1,19} \\ x_{2,1} & x_{2,2} & \cdots & x_{2,19} \\ \vdots & \vdots & \cdots & \vdots \\ x_{19,1} & x_{19,2} & \cdots & x_{19,19} \end{bmatrix}$$

4. Normalized influence matrix Q is obtained by dividing each entry of direct influence matrix X by the maximum value of the sum of the entries of each row of matrix X. That is,

$$Q = \frac{1}{d} \cdot X, d = \max_{1 \leq i \leq 19} \sum_{j=1}^{19} x_{ij}.$$

5. Calculate comprehensive influence matrix R, $R = Q(E-Q)^{-1}$, where E is the identity matrix.

6. Calculate the impact degree, influence degree, centrality degree, and cause degree. The sum of the elements in each row of comprehensive influence matrix R is called the impact degree of each factor, and the sum of the elements in each column is called the influence degree of each factor. Note that the impact degree is E, the influence degree is F, the centrality degree is H, and the cause degree is M. Here, $H = E+F$, $M = E-F$. If the cause degree is greater than 0, it is called a cause factor; if the cause degree is less than 0, it is called a result factor.

7. Draw a causality diagram and represent causality by a two-dimensional diagram with $(H_i, M_i)$ as the coordinates, with $H_i$ on the horizontal axis and $M_i$ on the vertical axis. The causality diagram enables decision makers to see the causal relationship between factors more intuitively, which is convenient for decision-making.

8. Calculate the overall impact matrix. Overall influence matrix $H(H = [h_{ij}]_{n \times n})$ is composed of comprehensive influence matrix R and identity matrix E, that is, $H = R+E$.

9. Determine the reachability matrix. When determining reachability matrix $K(K = [k_{ij}]_{n \times n})$, it is necessary to introduce threshold $\lambda$ to process the overall influence matrix. The formula is as follows:

$$k_{ij} = \begin{cases} 1, h_{ij} \geq \lambda(i,j = 1, 2, \cdots, n) \\ 0, h_{ij} < \lambda(i,j = 1, 2, \cdots, n) \end{cases}$$

The setting of threshold $\lambda$ eliminates the influence relationship between indicators with a small influence degree and simplifies the system structure. It facilitates subsequent hierarchical structure division. If the threshold value is larger, the independence between indicators is stronger, the system structure is simpler, and it is difficult to describe the influence relationship between indicators. If the threshold value is small, the influence relationship between indicators is more complex, and the integrity of the system is more difficult to express clearly. In this study, the method of Zhu et al. was adopted to select the threshold, and the threshold was verified by several value analysis experiments to obtain the best system structure model (Zhu et al. 2020).

10. Determine reachable set $R_i$ and antecedent set $S_i$, as follows:

$$\begin{cases} R_i = \{a_j | a_j \in A, k_{ij} = 1\}(i,j = 1, 2, \cdots, n) \\ S_i = \{a_i | a_i \in A, k_{ij} = 1\}(i,j = 1, 2, \cdots, n) \end{cases}$$

11. Determine the risk factor Ri of each layer in the hierarchical structure model as follows:

$$R_i = R_i \cap S_i(i = 1, 2, \cdots, n)$$

If the above equation is satisfied, then the antecedents of the elements in $R_i$ can be found in $S_i$, and $ai$ is the highest-level element. The $ith$ row and the $ith$ column are crossed out from reachable matrix $K$, and this step is repeated until all levels are determined. The influencing factor model of the cross-border e-commerce ecosystem is constructed based on the determined elements at different levels.

## 4. Identification of the key influencing factors of the cross-border e-commerce ecosystem

The cross-border e-commerce ecosystem is affected by multiple factors. This study seeks to more comprehensively identify the relevant factors affecting the cross-border e-commerce ecosystem. Initially, literature search was conducted on various academic search engines, such as Scopus, Web of Science, Google Scholar, and CNKI, using a combination of keywords such as "cross-border electronic commerce," "ecosystem," "influence mechanism of e-commerce ecosystem," and their combinations. Relevant literature from the last decade was selected, with the search timeframe ranging from December 2012 to December 2022. Core journal articles and highly cited papers related to the influencing factors of "cross-border e-commerce ecosystem" were then organized, summarized, and analysed. The main references for this study are references [2,11,39,40]. A typical scientific, comprehensive, and practical analysis of influencing factors was carried out and the results were refined and combined through expert interviews to form a preliminary indicator system. Subsequently, seven experts were invited to participate in the refinement of the preliminary indicator system. The panel was composed of three academic experts in the field of cross-border e-commerce (all holding the rank of associate professor or higher), two practitioners with over five years of experience in cross-border e-commerce, and two relevant government officials. The panel reviewed the initial theoretical indicators and made adjustments based on the following criteria: (1) if one expert suggested adding a specific indicator, the entire panel would discuss and determine whether to include it; (2) indicators deemed unimportant by two or more experts were deleted; and (3) in cases of disagreement, the panel would collectively discuss whether to modify or delete the indicator. Based on expert discussions and opinions, the influencing factors were divided into four primary factors: internal enterprise, policy support, economic foundation, and external environment, and subdivided into 19 specific influencing factors as shown in Table 1. The influencing factors are interrelated and jointly affect the cross-border e-commerce ecosystem. The analysis of the relationships between the influencing factors and the identification of the key influencing factors will help to provide a reference and decision-making support for the operation and management practice of the cross-border e-commerce ecosystem.

## 5. Establishment of an influencing factor model of the cross-border e-commerce ecosystem based on the DEMATEL-ISM method

### 5.1 Importance ranking of influencing factors and causal relationship analysis based on the DEMATEL method

This study primarily collected data through survey and interview methods. After conducting research on experts, teachers in the field of cross-border e-commerce, and leaders of management departments related to e-commerce, an expert team of 11 members was invited to participate in the study. In addition to the 3 experts from academic research in cross-border e-commerce (all holding associate professor or higher titles), 2 professionals with over 5 years of experience in cross-border e-commerce related industries, 2 relevant government officials mentioned in Chapter 4, and 4 cross-border e-commerce consumers who have been

**Table 1. Summary table of the influencing factors of the cross-border e-commerce ecosystem.**

| First-order factor | Specific influencing factor indicators | serial number |
|---|---|---|
| Internal enterprise factors | Development level of cross-border e-commerce platforms | S1 |
| | Development level of cross-border e-commerce logistics | S2 |
| | Comprehensive service level of foreign trade | S3 |
| | Degree of cross-border payment security | S4 |
| | Certification management level | S5 |
| | Competition and cooperation among enterprises | S6 |
| Government support | Government subsidies | S7 |
| | Intensity of tax incentives | S8 |
| | Governmental supervision degree | S9 |
| | Intensity of infrastructure investment | S10 |
| Economic foundation | GDP | S11 |
| | Per capita disposable income | S12 |
| | Living standard of residents | S13 |
| External environment | Level of informatization development | S14 |
| | Investment in information technology construction | S15 |
| | Scale of cross-border e-commerce transactions | S16 |
| | Competitiveness of cross-border e-commerce | S17 |
| | Demand for cross-border online shopping | S18 |
| | Cultural differences | S19 |

purchasing goods through cross-border e-commerce for more than 5 years were added to the team. The expert team used interview and Delphi methods to evaluate the interaction of factors, and the scoring system was based on a five-point scale ranging from 0 to 4 (0 indicating no influence, and 4 indicating very strong influence). The direct influence matrix of 19 influencing factors is obtained through expert scoring. MATLAB software was used to calculate the direct influence matrix based on steps (4)-(7), and the comprehensive influence matrix of the DEMATEL method is obtained in Table 2. The calculation results of the impact degree, influence degree, centrality degree and cause degree are shown in Table 3. The cause-effect analysis diagram is shown in Fig 2 below.

The results of the DEMATEL model are analysed as follows:

1.  Impact level. The top three factors are the cross-border e-commerce platform development level S1, cross-border e-commerce competitiveness S17 and the cross-border e-commerce transaction scale S16, and their impact degrees are 2.34, 2.13 and 1.96, respectively, indicating that these three factors have the greatest impact on the other influencing factors.

2.  Impact level. The top three factors are the information development level S14, GDP S11 and cross-border online shopping demand S18, and the degree of influence is 1.71, 1.66 and 1.63, respectively. The results indicate that these three factors are easily affected by the other factors.

3.  Centrality: The top three factors are the cross-border e-commerce platform development level S1, cross-border e-commerce competitiveness S17 and the cross-border e-commerce transaction scale S16, with centrality values of 3.78, 3.50 and 3.34, respectively. The magnitude of centrality indicates the critical degree of this factor in the cross-border e-commerce ecosystem. The greater the impact of this factor in the cross-border e-commerce ecosystem, the more worthy of attention it is.

**Table 2. Comprehensive impact matrix of the influencing factors of the cross-border e-commerce ecosystem.**

|     | S1 | S2 | S3 | S4 | S5 | S6 | S7 | S8 | S9 | S10 | S11 | S12 | S13 | S14 | S15 | S16 | S17 | S18 | S19 |
|-----|----|----|----|----|----|----|----|----|----|-----|-----|-----|-----|-----|-----|-----|-----|-----|-----|
| S1  | 0.08 | 0.13 | 0.11 | 0.12 | 0.11 | 0.10 | 0.11 | 0.14 | 0.13 | 0.13 | 0.13 | 0.12 | 0.13 | 0.15 | 0.13 | 0.13 | 0.13 | 0.15 | 0.11 |
| S2  | 0.12 | 0.05 | 0.08 | 0.07 | 0.06 | 0.08 | 0.09 | 0.09 | 0.11 | 0.12 | 0.10 | 0.10 | 0.10 | 0.12 | 0.10 | 0.10 | 0.10 | 0.12 | 0.07 |
| S3  | 0.10 | 0.10 | 0.04 | 0.09 | 0.08 | 0.09 | 0.09 | 0.10 | 0.09 | 0.09 | 0.10 | 0.08 | 0.09 | 0.11 | 0.09 | 0.10 | 0.09 | 0.11 | 0.08 |
| S4  | 0.10 | 0.07 | 0.05 | 0.04 | 0.07 | 0.05 | 0.06 | 0.07 | 0.08 | 0.08 | 0.08 | 0.07 | 0.08 | 0.11 | 0.09 | 0.09 | 0.09 | 0.10 | 0.07 |
| S5  | 0.05 | 0.04 | 0.04 | 0.05 | 0.03 | 0.04 | 0.05 | 0.06 | 0.07 | 0.05 | 0.07 | 0.06 | 0.07 | 0.09 | 0.07 | 0.07 | 0.07 | 0.07 | 0.05 |
| S6  | 0.12 | 0.09 | 0.09 | 0.10 | 0.09 | 0.04 | 0.07 | 0.10 | 0.08 | 0.09 | 0.10 | 0.08 | 0.09 | 0.12 | 0.10 | 0.11 | 0.11 | 0.12 | 0.09 |
| S7  | 0.04 | 0.03 | 0.03 | 0.03 | 0.03 | 0.04 | 0.03 | 0.06 | 0.04 | 0.05 | 0.08 | 0.06 | 0.07 | 0.05 | 0.06 | 0.05 | 0.05 | 0.06 | 0.06 |
| S8  | 0.05 | 0.03 | 0.03 | 0.03 | 0.03 | 0.04 | 0.06 | 0.03 | 0.04 | 0.05 | 0.08 | 0.06 | 0.07 | 0.05 | 0.05 | 0.05 | 0.05 | 0.06 | 0.06 |
| S9  | 0.06 | 0.05 | 0.05 | 0.06 | 0.06 | 0.05 | 0.05 | 0.09 | 0.04 | 0.07 | 0.07 | 0.06 | 0.08 | 0.09 | 0.06 | 0.08 | 0.08 | 0.07 | 0.08 |
| S10 | 0.07 | 0.04 | 0.03 | 0.04 | 0.03 | 0.04 | 0.07 | 0.06 | 0.05 | 0.03 | 0.09 | 0.06 | 0.07 | 0.07 | 0.06 | 0.05 | 0.05 | 0.07 | 0.07 |
| S11 | 0.06 | 0.04 | 0.04 | 0.05 | 0.04 | 0.04 | 0.07 | 0.09 | 0.05 | 0.06 | 0.05 | 0.09 | 0.08 | 0.09 | 0.07 | 0.06 | 0.05 | 0.07 | 0.05 |
| S12 | 0.04 | 0.03 | 0.03 | 0.03 | 0.03 | 0.04 | 0.04 | 0.07 | 0.04 | 0.04 | 0.09 | 0.03 | 0.08 | 0.06 | 0.05 | 0.04 | 0.03 | 0.06 | 0.05 |
| S13 | 0.05 | 0.04 | 0.03 | 0.03 | 0.03 | 0.04 | 0.04 | 0.07 | 0.04 | 0.05 | 0.09 | 0.09 | 0.03 | 0.06 | 0.05 | 0.04 | 0.04 | 0.06 | 0.06 |
| S14 | 0.05 | 0.04 | 0.03 | 0.06 | 0.05 | 0.04 | 0.05 | 0.06 | 0.06 | 0.07 | 0.08 | 0.07 | 0.08 | 0.04 | 0.10 | 0.05 | 0.05 | 0.08 | 0.07 |
| S15 | 0.09 | 0.05 | 0.05 | 0.06 | 0.06 | 0.05 | 0.07 | 0.07 | 0.06 | 0.08 | 0.10 | 0.08 | 0.08 | 0.09 | 0.04 | 0.07 | 0.06 | 0.09 | 0.08 |
| S16 | 0.12 | 0.08 | 0.09 | 0.11 | 0.09 | 0.08 | 0.07 | 0.13 | 0.08 | 0.11 | 0.11 | 0.12 | 0.13 | 0.12 | 0.10 | 0.06 | 0.12 | 0.13 | 0.11 |
| S17 | 0.13 | 0.12 | 0.11 | 0.12 | 0.10 | 0.09 | 0.11 | 0.14 | 0.12 | 0.11 | 0.12 | 0.11 | 0.10 | 0.14 | 0.12 | 0.11 | 0.07 | 0.13 | 0.09 |
| S18 | 0.07 | 0.05 | 0.06 | 0.06 | 0.06 | 0.05 | 0.08 | 0.11 | 0.06 | 0.07 | 0.10 | 0.10 | 0.11 | 0.10 | 0.08 | 0.09 | 0.08 | 0.05 | 0.09 |
| S19 | 0.03 | 0.03 | 0.01 | 0.01 | 0.01 | 0.02 | 0.02 | 0.01 | 0.02 | 0.01 | 0.02 | 0.01 | 0.02 | 0.03 | 0.02 | 0.03 | 0.03 | 0.02 | 0.01 |

Cause factor and result factor analysis. A factor whose cause degree is greater than 0 is called a cause factor. The influence factors of the ross-border e-commerce ecosystem are the cross-border electric business platform development level S1, the cross-border electronic business logistics development level S2, the foreign trade comprehensive service level S3, cross-

**Table 3. Impact degree results of the cross-border e-commerce ecosystem.**

|     | Impact degree | Degree of the affected | Concentricity | Centrality ranking | Cause degree | Factor attribute |
|-----|---------------|------------------------|---------------|--------------------|--------------|------------------|
| S1  | 2.34 | 1.44 | 3.78 | 1  | 0.89  | cause factor  |
| S2  | 1.78 | 1.10 | 2.88 | 5  | 0.67  | cause factor  |
| S3  | 1.71 | 1.01 | 2.72 | 10 | 0.70  | cause factor  |
| S4  | 1.49 | 1.15 | 2.64 | 11 | 0.34  | cause factor  |
| S5  | 1.09 | 1.07 | 2.16 | 17 | 0.02  | cause factor  |
| S6  | 1.82 | 1.03 | 2.85 | 7  | 0.79  | cause factor  |
| S7  | 0.92 | 1.24 | 2.16 | 18 | -0.33 | result factor |
| S8  | 0.92 | 1.53 | 2.45 | 14 | -0.61 | result factor |
| S9  | 1.26 | 1.26 | 2.52 | 12 | 0.00  | cause factor  |
| S10 | 1.06 | 1.38 | 2.43 | 15 | -0.32 | result factor |
| S11 | 1.15 | 1.66 | 2.81 | 8  | -0.51 | result factor |
| S12 | 0.88 | 1.47 | 2.35 | 16 | -0.59 | result factor |
| S13 | 0.94 | 1.58 | 2.51 | 13 | -0.64 | result factor |
| S14 | 1.14 | 1.71 | 2.86 | 6  | -0.57 | result factor |
| S15 | 1.34 | 1.43 | 2.77 | 9  | -0.10 | result factor |
| S16 | 1.96 | 1.38 | 3.34 | 3  | 0.59  | cause factor  |
| S17 | 2.13 | 1.38 | 3.50 | 2  | 0.75  | cause factor  |
| S18 | 1.48 | 1.63 | 3.11 | 4  | -0.15 | result factor |
| S19 | 0.40 | 1.35 | 1.75 | 19 | -0.96 | result factor |

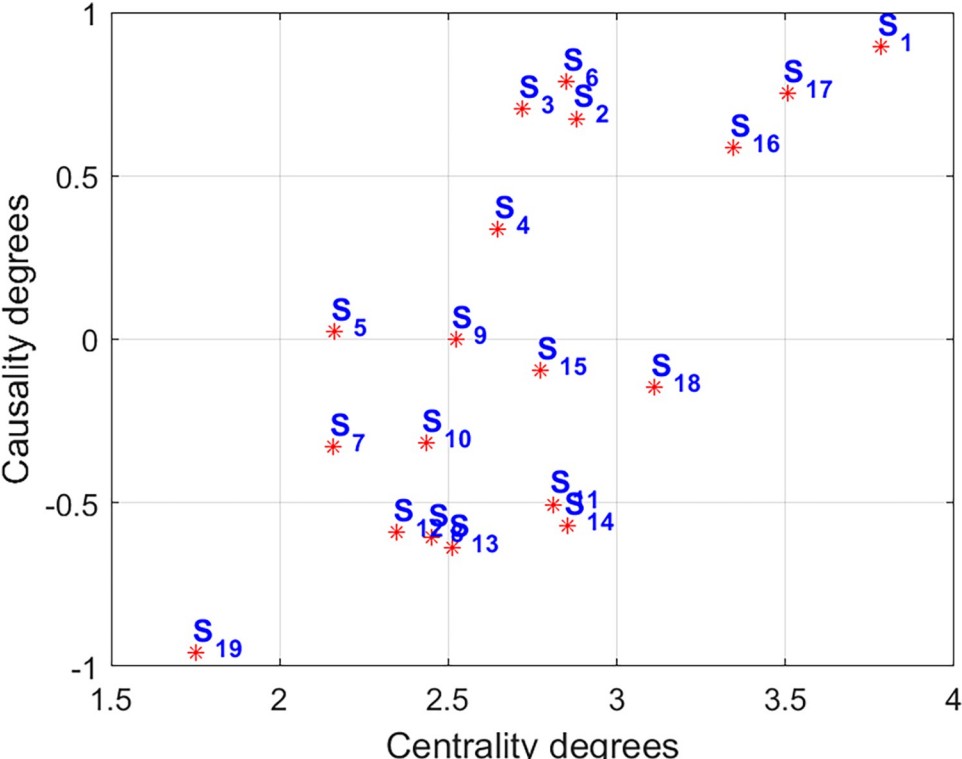

**Fig 2. Cause-effect analysis diagram.**

border payment security S4, the authentication management level S5, competition between enterprises S6, the integration of government regulation in cross-border e-commerce S16, and cross-border trade scale e-commerce competitiveness S17. These factors have a relatively large impact on the other factors. Among them, the level of cross-border e-commerce platform development S1, competition and cooperation among enterprises S6, and the competitiveness of cross-border e-commerce S17 are the main reasons. Factors with cause degrees less than 0 are called result factors, and they are easily affected by external factors. Among the influencing factors of the cross-border e-commerce ecosystem, the factors that are highly affected include government subsidy intensity S7, tax incentive intensity S8, infrastructure investment intensity S10, GDP S11, per capita disposable income S12, residents' living standard S13, the informatization development level S14, information technology construction investment S15, cross-border online shopping demand S18, and cultural differences S19.

## 5.2 Hierarchical structure analysis of influencing factors based on the ISM method

To reveal the influence mechanism between cause factors and result factors and determine the transmission path of the influence of key factors on the cross-border e-commerce ecosystem, based on equations (9)-(11) and the basic idea of dividing levels, to obtain the reachable matrix, it is necessary to set threshold λ. The setting of the λ value removes the influence relationship between indicators with a weak influence degree. The value of λ will affect the complexity of the whole system. The result for λ must be moderate. The average value of the comprehensive influence matrix is approximately 0.07, and the λ values are taken to be 0.05, 0.07, 0.09, 0.11, 0.13, 0.15 and 0.17 and are brought into the MATLAB program. According to

**Table 4. Reachability matrix of the influencing factors of the cross-border e-commerce ecosystem.**

|  | S1 | S2 | S3 | S4 | S5 | S6 | S7 | S8 | S9 | S10 | S11 | S12 | S13 | S14 | S15 | S16 | S17 | S18 | S19 |
|---|---|---|---|---|---|---|---|---|---|---|---|---|---|---|---|---|---|---|---|
| S1 | 1 | 1 | 0 | 1 | 1 | 0 | 1 | 1 | 1 | 1 | 1 | 1 | 1 | 1 | 1 | 1 | 1 | 1 | 0 |
| S2 | 1 | 1 | 0 | 0 | 0 | 0 | 0 | 0 | 0 | 1 | 0 | 0 | 0 | 1 | 0 | 0 | 0 | 1 | 0 |
| S3 | 0 | 0 | 1 | 0 | 0 | 0 | 0 | 0 | 0 | 0 | 0 | 0 | 0 | 1 | 0 | 0 | 0 | 0 | 0 |
| S4 | 0 | 0 | 0 | 1 | 0 | 0 | 0 | 0 | 0 | 0 | 0 | 0 | 0 | 1 | 0 | 0 | 0 | 0 | 0 |
| S5 | 0 | 0 | 0 | 0 | 1 | 0 | 0 | 0 | 0 | 0 | 0 | 0 | 0 | 0 | 0 | 0 | 0 | 0 | 0 |
| S6 | 1 | 0 | 0 | 0 | 0 | 1 | 0 | 0 | 0 | 0 | 0 | 0 | 0 | 1 | 0 | 1 | 1 | 1 | 0 |
| S7 | 0 | 0 | 0 | 0 | 0 | 0 | 1 | 0 | 0 | 0 | 0 | 0 | 0 | 0 | 0 | 0 | 0 | 0 | 0 |
| S8 | 0 | 0 | 0 | 0 | 0 | 0 | 0 | 1 | 0 | 0 | 0 | 0 | 0 | 0 | 0 | 0 | 0 | 0 | 0 |
| S9 | 0 | 0 | 0 | 0 | 0 | 0 | 0 | 0 | 1 | 0 | 0 | 0 | 0 | 0 | 0 | 0 | 0 | 0 | 0 |
| S10 | 0 | 0 | 0 | 0 | 0 | 0 | 0 | 0 | 0 | 1 | 0 | 0 | 0 | 0 | 0 | 0 | 0 | 0 | 0 |
| S11 | 0 | 0 | 0 | 0 | 0 | 0 | 0 | 0 | 0 | 0 | 1 | 0 | 0 | 0 | 0 | 0 | 0 | 0 | 0 |
| S12 | 0 | 0 | 0 | 0 | 0 | 0 | 0 | 0 | 0 | 0 | 0 | 1 | 0 | 0 | 0 | 0 | 0 | 0 | 0 |
| S13 | 0 | 0 | 0 | 0 | 0 | 0 | 0 | 0 | 0 | 0 | 0 | 0 | 1 | 0 | 0 | 0 | 0 | 0 | 0 |
| S14 | 0 | 0 | 0 | 0 | 0 | 0 | 0 | 0 | 0 | 0 | 0 | 0 | 0 | 1 | 0 | 0 | 0 | 0 | 0 |
| S15 | 0 | 0 | 0 | 0 | 0 | 0 | 0 | 0 | 0 | 0 | 0 | 0 | 0 | 0 | 1 | 0 | 0 | 0 | 0 |
| S16 | 1 | 0 | 0 | 0 | 0 | 0 | 0 | 1 | 0 | 0 | 1 | 1 | 1 | 1 | 0 | 1 | 1 | 1 | 0 |
| S17 | 1 | 1 | 1 | 1 | 0 | 0 | 1 | 1 | 1 | 1 | 1 | 0 | 0 | 1 | 1 | 1 | 1 | 1 | 0 |
| S18 | 0 | 0 | 0 | 0 | 0 | 0 | 0 | 0 | 0 | 0 | 0 | 0 | 0 | 0 | 0 | 0 | 0 | 1 | 0 |
| S19 | 0 | 0 | 0 | 0 | 0 | 0 | 0 | 0 | 0 | 0 | 0 | 0 | 0 | 0 | 0 | 0 | 0 | 0 | 1 |

the results of repeated trials, the value of λ is 0.11, which is suitable. The reachability matrix of the cross-border e-commerce ecosystem is shown in Table 4 below. Additionally, the multi-level hierarchical structure model is shown in Fig 3.

According to the multi-level hierarchical structure relationship model of influencing factors, the role of each influencing factor in the system and its corresponding level can be clearly known from the figure. In the index set of the influencing factors of the cross-border e-commerce ecosystem, all factors are divided into four levels, and the degree of influence increases

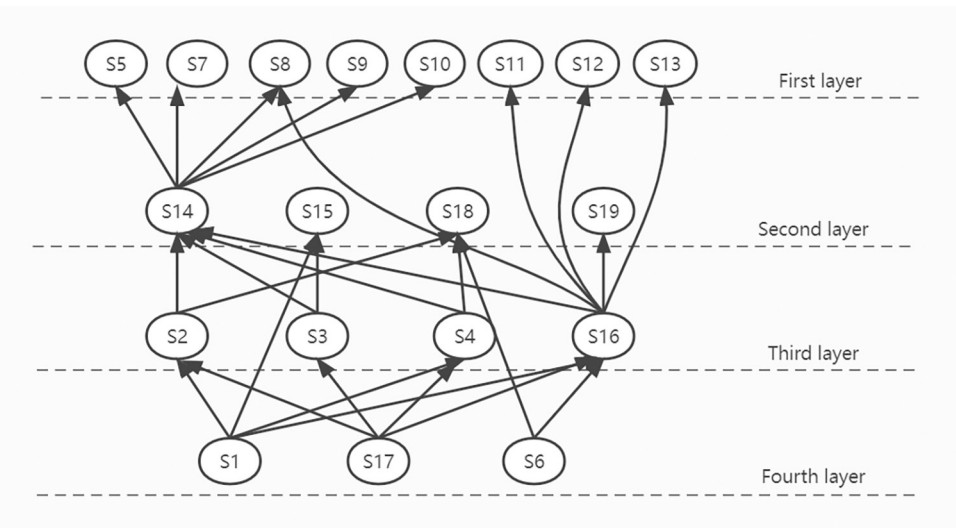

**Fig 3. Multi-level hierarchical structure model of the influencing factors of the cross-border e-commerce ecosystem.**

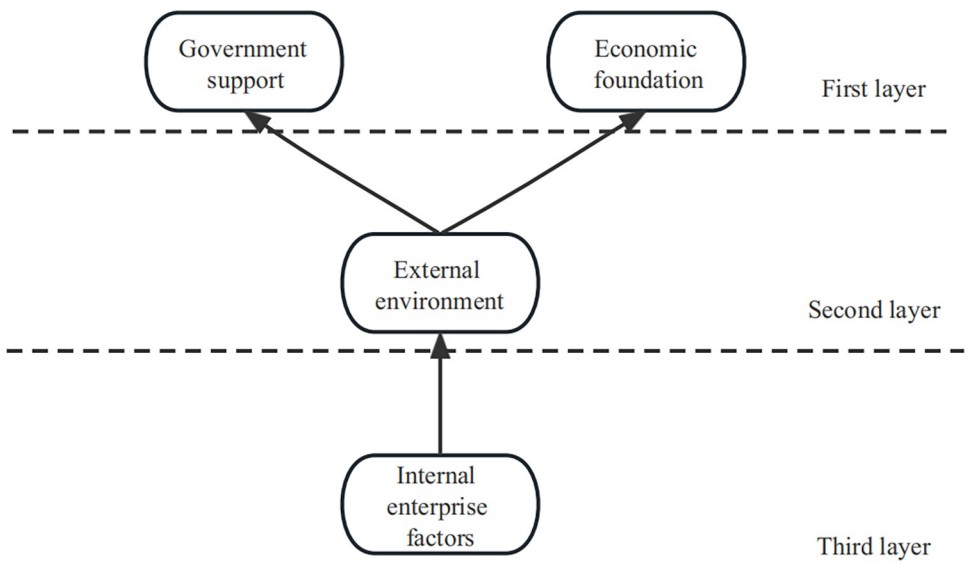

**Fig 4. Multi-level hierarchical structure model of first-level influencing factors of cross-border e-commerce ecosystem.**

with increasing levels. As shown in the figure on the right, the first layer is the surface factor, including the certification management level (S5), the government subsidy level (S7), the tax incentive level (S8), the government supervision level (S9), the infrastructure investment level (S10), GDP (S11), per capita disposable income (S12), and residents' living standard (S13), which are the direct factors affecting the e-commerce ecosystem. The level of cross-border e-commerce platform development (S1), the competitiveness of cross-border e-commerce (S17), and the competition and cooperation between enterprises (S6) are the root factors affecting the cross-border e-commerce ecosystem, which should be paid attention to. The eight factors located in the second to third layers connect the direct factors and the root factors, connecting the factor system.

Based on the multi-level hierarchical structure model of the influencing factors of cross-border e-commerce ecosystem shown in Fig 3, we construct a multi-level hierarchical structure model of first-level influencing factors. Examining the superficial layer of Fig 3, we can see that the influencing factor indicators mostly belong to the first-level indicators of policy support and economic foundation. Therefore, we assign policy support and economic foundation to the first layer. The second layer of Fig 3 includes influencing factors that mostly belong to the first-level indicator of external environment; hence, we group external environment into the second layer. Similarly, the third and fourth layers of Fig 3 consist mostly of influencing factors belonging to the first-level indicator of internal enterprise, so we assign this indicator to the third layer of Fig 4. Following this logical pattern, we can construct a multi-level hierarchical structure model of first-level influencing factors, as shown in Fig 4.

### 5.3 Comprehensive analysis

Combining the comprehensive analysis of the results of the DEMATEL-ISM method, the impact degree, influence degree, centrality degree and cause degree of 19 factors of the cross-border e-commerce ecosystem are calculated by the DEMATEL method. After an in-depth analysis of the indicators, it is found that the e-commerce platform development level (S1), cross-border e-commerce competitiveness (S17) and the cross-border e-commerce transaction

scale (S16) have the greatest impact on the other influencing factors. Additionally, the informatization development level (S14), GDP (S11) and cross-border online shopping demand (S18) are easily affected by the other factors. As root factors, the level of cross-border e-commerce platform development (S1), the competitiveness of cross-border e-commerce (S17) and competition and cooperation among enterprises (S6) play an important role in the influencing factors of the cross-border e-commerce ecosystem, which will affect the stability of the cross-border e-commerce ecosystem by affecting the other factors.

### 5.4 Mutual verification of DEMATEL and ISM modeling results

The DEMATEL method identifies key elements and their degree of influence in complex systems through measures such as centrality and causality. On the other hand, the ISM method determines the inherent logical structure and hierarchy of elements in a system. By combining the strengths of both algorithms, this study creates a hierarchical and structured model with greater explanatory power.

Comparing the results of the DEMATEL (Fig 2) and ISM (Fig 3) modeling, we can find correlations and to some extent, consistency between the analysis results of the two methods, thus verifying the accuracy of the model analysis. In the DEMATEL analysis, the factors with the highest influence degree on the development of e-commerce platforms (S1), cross-border e-commerce competitiveness (S17), and cross-border e-commerce transaction scale (S16) belong to the deep-layer factors in the ISM analysis. Conversely, factors with lower influence degrees belong to surface-level factors in the ISM analysis. Moreover, the root-layer influencing factors in the multi-level hierarchical structure model, namely the development of e-commerce platforms (S1), cross-border e-commerce competitiveness (S17), and the competition and cooperation between enterprises (S6), correspond to the set of reasons. By comparing the graphical results, we find that the two models have consistency in identifying the importance and type of influencing factors in the cross-border e-commerce ecosystem, thereby proving the effectiveness and feasibility of the model established in this study.

## 6. Conclusion and prospects

### 6.1 Management insights

Based on the aforementioned data analysis results, the following management insights can be proposed:

1.  There is a need to optimize the construction of China's cross-border e-commerce platforms and logistics services. In addition to strengthening interaction and communication with foreign consumers to promote trust, emphasis should also be placed on optimizing product structure and innovation, enhancing quality control, and improving the international competitiveness of cross-border e-commerce to meet the ever-changing demands of consumers. At the same time, it is necessary to promote the development of cross-border logistics, strengthen infrastructure construction such as roads, railways, overseas warehouses, logistics information systems, etc., optimize business and services, enhance consumer experience, and thereby promote the increase in cross-border e-commerce platform sales volume, and further promote the healthy development of the cross-border e-commerce ecosystem.

2.  Attention should be paid to the cultivation of relevant cross-border e-commerce talents, the establishment of a sound talent training system and innovation and entrepreneurship support system, and the provision of sufficient reserve talents for the development of cross-border e-commerce. The government, enterprises, and universities should work together to train talents, use advanced information technology, and build an information support

platform. The government should also continue to play a role in promoting the development of cross-border e-commerce, and implement relevant support policies, increase regulatory efforts, and build cross-border e-commerce infrastructure and other measures to create a good market, policy, legal, credit, technology, and other environment for the healthy and stable development of the cross-border e-commerce ecosystem.

3. The government needs to take the lead in promoting cooperation among e-commerce-related enterprises, and increasing opportunities for cooperation and communication across regions and different countries. There is also a need for expanding the scale of cross-border e-commerce transactions, enhancing the core competitiveness of China's cross-border e-commerce, and effectively promoting the further progress and development of the cross-border e-commerce ecosystem.

## 6.2 Research contributions

This study has multiple contributions. First, based on the previous work, this study used literature mining and expert interviews to identify factors that affect the development of the cross-border e-commerce ecosystem and determine key factors. Second, this study proposed the DEMATEL-ISM model to study the interaction between factors and provide insights and guidance for cross-border e-commerce management decision-makers. At the same time, this study applied the DEMATEL-ISM model to the cross-border e-commerce ecosystem, providing a reference research paradigm for related studies. In addition, the author hopes that the findings of this study can be used as a reference for the operation and management of the cross-border e-commerce ecosystem. It is hoped that the key indicators affecting the cross-border e-commerce ecosystem identified by this quantitative method can provide important references and decision-making support for its operation and management.

The findings of this study have certain reference significance for cross-border e-commerce ecosystem operation and management. On the basis of a literature review and full investigation, this study identified 19 factors affecting the cross-border e-commerce ecosystem. The importance and correlation of related factors were further analysed. It is hoped that the key indicators affecting the cross-border e-commerce ecosystem identified by this quantitative method can provide an important reference and decision support for its operation management.

## 6.3 Research prospects

This study still has some areas that can be expanded, and future authors may focus on the following aspects:

1. Exploring the driving and diffusion mechanisms of the cross-border e-commerce ecosystem
   Identify the evolutionary paths that drive the cross-border e-commerce ecosystem. Based on the different periods that drive the evolution of the cross-border e-commerce ecosystem, the similarities and differences of its driving and diffusion mechanisms are distinguished. Through different data collection and analysis methods, a scientific and reasonable management and decision model can be established to analyse how to drive the evolution of the cross-border e-commerce ecosystem.

2. Propose a universal cross-border e-commerce ecosystem model and implementation path
   Various models of cross-border e-commerce ecosystems have been proposed in the literature. However, for different types of enterprises, how should they participate in cross-

border e-commerce ecosystem management? In addition, based on different cultural and policy backgrounds, the establishment of a reasonable ecosystem return model is still the focus and challenge of future research.

## Supporting information

**S1 Dataset.**
(XLSX)

**S1 File. Questionnaire file.**
(ZIP)

**S2 File. Expert information file.**
(DOCX)

**S3 File. Calculation code file.**
(ZIP)

## Acknowledgments

The authors would like to express their kindest gratitude to survey respondents and anonymous reviewers. The experts consented to having their identity published, and the authors promise that the relevant information will be used solely for academic research purposes.

## Author Contributions

**Supervision:** Mingxia Wei, Brian Sheng-Xian Teo.

**Writing – original draft:** Xiaodan Xi.

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
