## [Decision Letter · Decision Letter 0]

28 Mar 2023

PONE-D-23-03573Analysis of the Key Influencing Factors of a Cross-Border E-Commerce Ecosystem Based on the DEMATEL-ISM MethodPLOS ONE

Dear Dr. Xi,

Thank you for submitting your manuscript to PLOS ONE. After careful consideration, we feel that it has merit but does not fully meet PLOS ONE’s publication criteria as it currently stands. Therefore, we invite you to submit a revised version of the manuscript that addresses the points raised during the review process.

We look forward to receiving your revised manuscript.

Kind regards,

Tinggui Chen

Academic Editor

PLOS ONE

Journal Requirements:

Additional Editor Comments:

I have completed my evaluation of your manuscript. The reviewers recommend reconsideration of your manuscript following major revision. I invite you to resubmit your manuscript after addressing the comments.

Reviewers' comments:

Reviewer's Responses to Questions

**Comments to the Author**

1. Is the manuscript technically sound, and do the data support the conclusions?

Reviewer #1: Yes

Reviewer #2: Yes

2. Has the statistical analysis been performed appropriately and rigorously? 

Reviewer #1: Yes

Reviewer #2: I Don't Know

3. Have the authors made all data underlying the findings in their manuscript fully available?

Reviewer #1: No

Reviewer #2: No

4. Is the manuscript presented in an intelligible fashion and written in standard English?

Reviewer #1: No

Reviewer #2: Yes

5. Review Comments to the Author

Reviewer #1: This paper studies many factors affecting the cross-border e-commerce ecosystem, which has certain innovative value.

1. From the text description of the research background, this paper conducts research based on the actual trade and cross-border e-commerce development in China, but the research title does not show this feature, so the title may need to be changed? Otherwise, it raises questions about whether China's cross-border e-commerce is representative of the world.

2. Conclusion and prospect of the sixth part. The conclusion part is very weak and needs to be elaborated. At the same time, the enlightenment and application value of the conclusion in management should also be elaborated, otherwise the application value of this study will be lost.

3. The source of the research data has not been explained clearly. When was it carried out? The paper says "search on the website", what is the time node of search?

4. The research method of expert interview is adopted in this paper. What kind of experts were interviewed and what role did the experts play in the study?

5. The overall number of references is relatively small, and there are more Chinese references. In order to make your paper more communicated with international researchers, this situation should be changed.

Reviewer #2: The paper presents the certain reference significance for cross-border e-commerce

ecosystem operation and management. On the basis of a literature review and full investigation,

this study identified 19 factors affecting the cross-border e-commerce ecosystem. The stud is interesting for researchers and practitioners. However, a number of improvements are needed.

1. Please provide the weakness in the related work.

2. Why and how does this study use many methods?

3. Please provide the profile of the experts in the study. how do you get the data?

4. Please provide the Multi-level hierarchical structure model of the influencing the first-order factors of the cross-border e-commerce ecosystem

5. Please compare the results of fig. 2 and Fig 3.

6. Please identify the contribution of this study.

7. Please update the reference.

6. PLOS authors have the option to publish the peer review history of their article (what does this mean?). If published, this will include your full peer review and any attached files.

Reviewer #1: No

Reviewer #2: No

---

## [Author Response · Author response to Decision Letter 0]

26 Apr 2023

Response to Reviewers

Dear Chen

Title：Analysis of the key influencing factors of a China's cross-border e-commerce ecosystem based on the DEMATEL-ISM method

Authors：Xiaodan Xi, Mingxia Wei, Brian Sheng-Xian Teo

Dear Tinggui Chen,

We would like to thank you for the prompt review of our paper and for the opportunity to respond to Academic Editor, Reviewers #1, #2. All reviewers provided us with insightful and valuable comments. We have revised our paper and addressed all of the issues raised by all reviewers, based on their comments and suggestions. We have attached a point by point response for the reviewers. Please see the details bellow. Thank you again for giving us the opportunity to revise the paper. Suggestions and comments from the reviewers have further significantly improved our paper.

Sincerely,

The Authors

Encl.

(1) Responses to comments of Academic Editor

(2) Responses to comments of Reviewer #1

(3) Responses to comments of Reviewer #2

Responses to Comments of Academic Editor

We would like to thank you for your time spent in reviewing our paper and for providing us with valuable comments and suggestions. We also appreciate your positive view on our results and hope that this version has fully addressed your concerns. To make it easy to follow, we first show (in italic) your comment and then present our response (in normal font).

Comments

2. In your Data Availability statement, you have not specified where the minimal data set underlying the results described in your manuscript can be found. PLOS defines a study's minimal data set as the underlying data used to reach the conclusions drawn in the manuscript and any additional data required to replicate the reported study findings in their entirety. All PLOS journals require that the minimal data set be made fully available. 

3. PLOS requires an ORCID iD for the corresponding author in Editorial Manager on papers submitted after December 6th, 2016. Please ensure that you have an ORCID iD and that it is validated in Editorial Manager. To do this, go to ‘Update my Information’ (in the upper left-hand corner of the main menu), and click on the Fetch/Validate link next to the ORCID field.

Response/Revision: Firstly, we have made modifications to meets PLOS ONE's style requirements, including those for file naming. We have further uploaded the minimal data set related to the literature, which includes other relevant data such as questionnaire data and expert information. The corresponding authors have also registered their ORCID iD, which has been validated in Editorial Manager.

Thank you again for the suggestions and comments, which have further improved our paper.

Responses to Comments of Reviewer #1

We would like to thank you for your time spent in reviewing our paper and for providing us with valuable comments and suggestions. We also appreciate your positive view on our results and hope that this version has fully addressed your concerns. To make it easy to follow, we first show (in italic) your comment and then present our response (in normal font).

Comments

Comment #1 From the text description of the research background, this paper conducts research based on the actual trade and cross-border e-commerce development in China, but the research title does not show this feature, so the title may need to be changed? Otherwise, it raises questions about whether China's cross-border e-commerce is representative of the world.

Response/Revision: Thanks for the comment, and it is a very correct suggestion.In this revision, I've changed the title of the article to "Analysis of the key influencing factors of China's cross-border e-commerce ecosystem based on the DEMATEL-ISM method".

Comment #2 Conclusion and prospect of the sixth part. The conclusion part is very weak and needs to be elaborated. At the same time, the enlightenment and application value of the conclusion in management should also be elaborated, otherwise the application value of this study will be lost.

Response/Revision: Thanks for pointing out! Following your suggestion, In Section 5.4, we have enriched the conclusion of our study by adding a comparison of the results obtained from DEMATEL and ISM models, as evidenced by Figures 2 and 3 (In lines 423-444). In Section 6, we have elaborated on the management insights of our findings, as well as the research contributions of this paper, in three points. Finally, we have included a research prospects, and emphasized the practical value of this study (In lines 445-515). Please see the details bellow.

“5.4 Mutual verification of DEMATEL and ISM modeling results

The DEMATEL method identifies key elements and their degree of influence in complex systems through measures such as centrality and causality. On the other hand, the ISM method determines the inherent logical structure and hierarchy of elements in a system. By combining the strengths of both algorithms, this study creates a hierarchical and structured model with greater explanatory power.

Comparing the results of the DEMATEL (Fig 2) and ISM (Fig 3) modeling, we can find correlations and to some extent, consistency between the analysis results of the two methods, thus verifying the accuracy of the model analysis. In the DEMATEL analysis, the factors with the highest influence degree on the development of e-commerce platforms (S1), cross-border e-commerce competitiveness (S17), and cross-border e-commerce transaction scale (S16) belong to the deep-layer factors in the ISM analysis. Conversely, factors with lower influence degrees belong to surface-level factors in the ISM analysis. Moreover, the root-layer influencing factors in the multi-level hierarchical structure model, namely the development of e-commerce platforms (S1), cross-border e-commerce competitiveness (S17), and the competition and cooperation between enterprises (S6), correspond to the set of reasons. By comparing the graphical results, we find that the two models have consistency in identifying the importance and type of influencing factors in the cross-border e-commerce ecosystem, thereby proving the effectiveness and feasibility of the model established in this study.

6. Conclusion and Prospects

6.1 Management Insights

Based on the aforementioned data analysis results, the following management insights can be proposed:

(1)There is a need to optimize the construction of China's cross-border e-commerce platforms and logistics services. In addition to strengthening interaction and communication with foreign consumers to promote trust, emphasis should also be placed on optimizing product structure and innovation, enhancing quality control, and improving the international competitiveness of cross-border e-commerce to meet the ever-changing demands of consumers. At the same time, it is necessary to promote the development of cross-border logistics, strengthen infrastructure construction such as roads, railways, overseas warehouses, logistics information systems, etc., optimize business and services, enhance consumer experience, and thereby promote the increase in cross-border e-commerce platform sales volume, and further promote the healthy development of the cross-border e-commerce ecosystem.

(2) Attention should be paid to the cultivation of relevant cross-border e-commerce talents, the establishment of a sound talent training system and innovation and entrepreneurship support system, and the provision of sufficient reserve talents for the development of cross-border e-commerce. The government, enterprises, and universities should work together to train talents, use advanced information technology, and build an information support platform. The government should also continue to play a role in promoting the development of cross-border e-commerce, and implement relevant support policies, increase regulatory efforts, and build cross-border e-commerce infrastructure and other measures to create a good market, policy, legal, credit, technology, and other environment for the healthy and stable development of the cross-border e-commerce ecosystem.

(3) The government needs to take the lead in promoting cooperation among e-commerce-related enterprises, and increasing opportunities for cooperation and communication across regions and different countries. There is also a need for expanding the scale of cross-border e-commerce transactions, enhancing the core competitiveness of China's cross-border e-commerce, and effectively promoting the further progress and development of the cross-border e-commerce ecosystem.

6.2 Research Contributions

This study has multiple contributions. First, based on the previous work, this study used literature mining and expert interviews to identify factors that affect the development of the cross-border e-commerce ecosystem and determine key factors. Second, this study proposed the DEMATEL-ISM model to study the interaction between factors and provide insights and guidance for cross-border e-commerce management decision-makers. At the same time, this study applied the DEMATEL-ISM model to the cross-border e-commerce ecosystem, providing a reference research paradigm for related studies. In addition, the author hopes that the findings of this study can be used as a reference for the operation and management of the cross-border e-commerce ecosystem. It is hoped that the key indicators affecting the cross-border e-commerce ecosystem identified by this quantitative method can provide important references and decision-making support for its operation and management.

The findings of this study have certain reference significance for cross-border e-commerce ecosystem operation and management. On the basis of a literature review and full investigation, this study identified 19 factors affecting the cross-border e-commerce ecosystem. The importance and correlation of related factors were further analysed. It is hoped that the key indicators affecting the cross-border e-commerce ecosystem identified by this quantitative method can provide an important reference and decision support for its operation management.

6.3 Research Prospects

This study still has some areas that can be expanded, and future authors may focus on the following aspects:

(1) Exploring the driving and diffusion mechanisms of the cross-border e-commerce ecosystem

Identify the evolutionary paths that drive the cross-border e-commerce ecosystem. Based on the different periods that drive the evolution of the cross-border e-commerce ecosystem, the similarities and differences of its driving and diffusion mechanisms are distinguished. Through different data collection and analysis methods, a scientific and reasonable management and decision model can be established to analyse how to drive the evolution of the cross-border e-commerce ecosystem.

(2) Propose a universal cross-border e-commerce ecosystem model and implementation path

Various models of cross-border e-commerce ecosystems have been proposed in the literature. However, for different types of enterprises, how should they participate in cross-border e-commerce ecosystem management? In addition, based on different cultural and policy backgrounds, the establishment of a reasonable ecosystem return model is still the focus and challenge of future research.”

Comment #3 The source of the research data has not been explained clearly. When was it carried out? The paper says "search on the website", what is the time node of search?

Response: Thanks for your comments！In the fourth section of our article, we have added relevant descriptions (In lines 256-263). Our literature sources were obtained from research websites such as Scopus, Web of Science, Google Scholar, and CNKI, with search dates ranging from December 2012 to December 2022. In the supporting information section, we have included detailed profiles of relevant experts and questionnaire results, and we have described in detail the process by which the experts were involved in selecting the indicators. In section 5.1, we have further described the assessment of the interaction between the indicators by the members of the expert group (In lines 267-279). Please see the details bellow. 

“Initially, literature search was conducted on various academic search engines, such as Scopus, Web of Science, Google Scholar, and CNKI, using a combination of keywords such as "cross-border electronic commerce," "ecosystem," "influence mechanism of e-commerce ecosystem," and their combinations. Relevant literature from the last decade was selected, with the search timeframe ranging from December 2012 to December 2022. Core journal articles and highly cited papers related to the influencing factors of "cross-border e-commerce ecosystem" were then organized, summarized, and analysed. The main references for this study are references [2,11,39, and 40]. A typical scientific, comprehensive, and practical analysis of influencing factors was carried out and the results were refined and combined through expert interviews to form a preliminary indicator system. Subsequently, seven experts were invited to participate in the refinement of the preliminary indicator system. The panel was composed of three academic experts in the field of cross-border e-commerce (all holding the rank of associate professor or higher), two practitioners with over five years of experience in cross-border e-commerce, and two relevant government officials. The panel reviewed the initial theoretical indicators and made adjustments based on the following criteria: (1) if one expert suggested adding a specific indicator, the entire panel would discuss and determine whether to include it; (2) indicators deemed unimportant by two or more experts were deleted; and (3) in cases of disagreement, the panel would collectively discuss whether to modify or delete the indicator. Based on expert discussions and opinions, the influencing factors were divided into four primary factors: internal enterprise, policy support, economic foundation, and external environment, and subdivided into 19 specific influencing factors as shown in Table 1.

5.1 Importance ranking of influencing factors and causal relationship analysis based on the DEMATEL method

This study primarily collected data through survey and interview methods. After conducting research on experts, teachers in the field of cross-border e-commerce, and leaders of management departments related to e-commerce, an expert team of 11 members was invited to participate in the study. In addition to the 3 experts from academic research in cross-border e-commerce (all holding associate professor or higher titles), 2 professionals with over 5 years of experience in cross-border e-commerce related industries, 2 relevant government officials mentioned in Chapter 4, and 4 cross-border e-commerce consumers who have been purchasing goods through cross-border e-commerce for more than 5 years were added to the team. The expert team used interview and Delphi methods to evaluate the interaction of factors, and the scoring system was based on a five-point scale ranging from 0 to 4 (0 indicating no influence, and 4 indicating very strong influence).”

Comment #4 The research method of expert interview is adopted in this paper. What kind of experts were interviewed and what role did the experts play in the study?

Response/Revision: Thanks for comments! Firstly, in the supporting information section, we have uploaded detailed profiles of the relevant experts and the results of the questionnaire data. Furthermore, in the fourth section, we have provided a detailed description of the expert's involvement in the selection of indicators (In lines 267-279). Additionally, in section 5.1, we have described the expert group's evaluation of the interactions among the indicators (In lines 293-304). These interviewees are authoritative figures in the field, with a profound understanding and insights into the e-commerce ecosystem, providing suggestions with depth and breadth. In the interviews and questionnaire surveys, the role of the experts is to answer questions, share experiences, provide insights and perspectives, to help researchers obtain more information about the e-commerce ecosystem theme. Please see the details bellow.

“Subsequently, seven experts were invited to participate in the refinement of the preliminary indicator system. The panel was composed of three academic experts in the field of cross-border e-commerce (all holding the rank of associate professor or higher), two practitioners with over five years of experience in cross-border e-commerce, and two relevant government officials. The panel reviewed the initial theoretical indicators and made adjustments based on the following criteria: (1) if one expert suggested adding a specific indicator, the entire panel would discuss and determine whether to include it; (2) indicators deemed unimportant by two or more experts were deleted; and (3) in cases of disagreement, the panel would collectively discuss whether to modify or delete the indicator. Based on expert discussions and opinions, the influencing factors were divided into four primary factors: internal enterprise, policy support, economic foundation, and external environment, and subdivided into 19 specific influencing factors as shown in Table 1. 

5.1 Importance ranking of influencing factors and causal relationship analysis based on the DEMATEL method

This study primarily collected data through survey and interview methods. After conducting research on experts, teachers in the field of cross-border e-commerce, and leaders of management departments related to e-commerce, an expert team of 11 members was invited to participate in the study. In addition to the 3 experts from academic research in cross-border e-commerce (all holding associate professor or higher titles), 2 professionals with over 5 years of experience in cross-border e-commerce related industries, 2 relevant government officials mentioned in Chapter 4, and 4 cross-border e-commerce consumers who have been purchasing goods through cross-border e-commerce for more than 5 years were added to the team. The expert team used interview and Delphi methods to evaluate the interaction of factors, and the scoring system was based on a five-point scale ranging from 0 to 4 (0 indicating no influence, and 4 indicating very strong influence).”

Comment #5 The overall number of references is relatively small, and there are more Chinese references. In order to make your paper more communicated with international researchers, this situation should be changed.

Response/Revision: Thanks for the comment! This is a very good suggestion. In order to make my paper more communicated with international researchers, we have reorganized the reference section and added English literature closely related to the research, expanding the number of references to 40. Please see the details bellow.

“References

1.Cumming D, Johan S, Khan Z, Meyer M. E-Commerce Policy and International Business. Management international review : MIR : journal of international business. 2022; 63(1): 3-25.

2.Gao TG. Study on the Intention of Foreign Trade Driven by Cross-Border E-Commerce Based on Blockchain Technology. Security and Communication Networks. 2021.

3.Zhang XH. Construction mechanism and implementation path of cross-border e-commerce ecosystem. Contemporary Economic Management. 2021; 43(07): 55-60.

4.Sun LB, Lyu GD, Yu YG, Teo CP. Cross-Border E-commerce Data Set: Choosing the Right Fulfillment Option. Manufacturing & Service Operations Management. 2020; 23(5): 1297-1313.

5.Deng XG, Ouyang YX. Cross-Border Supply Chain System Constructed by Complex Computer Blockchain for International Cooperation. Computational intelligence and neuroscience. 2022.

6.Xue CG, Zhou ML,Cao WJ. Research on dynamic mechanism of cross-border e-commerce ecosystem based on system dynamics. Industrial engineering. 2020; 23(04): 84-92.

7.Zhu JS, Lan WD, Zhang XC. Geographic proximity, supply chain and organizational glocalized survival: China's e-commerce investments in Indonesia. PloS one. 2021; 16(9).

8.Miao M, Krishna J. Mobile payments in Japan, South Korea and China: Cross-border convergence or divergence of business models?. Telecommunications Policy. 2016; 40(2-3):182-196

9.Tikhomirova A, Huang J, Chuanmin S, Khayyam M, Ali H, Khramchenko DS. How Culture and Trustworthiness Interact in Different E-Commerce Contexts: A Comparative Analysis of Consumers' Intention to Purchase on Platforms of Different Origins. Frontiers in Psychology, 2021,12.

10.Du J,Yu ZY. Building a Cross-Border E-Commerce Ecosystem Model Based on Block Chain + Internet of Things. Security and Communication Networks. 2021.

11.He J, Li JJ, Ge L. Model and Simulation of Symbiotic Evolutionary Dynamics of a Marine Cross-Border E-Commerce Trade Ecosystem. JOURNAL OF COASTAL RESEARCH. 2020; 108: 95-98.

12.Wulfert T,Woroch R, Strobel G, Seufert S, Möller F. Developing design principles to standardize e-commerce ecosystems: A systematic literature review and multi-case study of boundary resources. Electronic markets. 2022; 32(4): 1813-1842.

13.Rong K, Zhou D, Shi XW, Huang W. Social Information Disclosure of Friends in Common in an E‐commerce Platform Ecosystem: An Online Experiment. Production and Operations Management. 2021; 31(3): 984-1005.

14.Zhang XH. Jingdong: Build a cross-border e-commerce ecosystem. Enterprise management. 2016; (11):102-104.

15.Peng C, Jing X,Tie J, Tian Y, Kong JY, Xue K, Zhou Y. Research on Value Co-Creation New Business Model of Import Cross-Border E-Commerce Platform Ecosystem. Security and Communication Networks. 2022.

16.You J, Peng LH.Study on ecological characteristics of e-commerce ecosystem. Enterprise economic. 2017; 36(08):115-122.

17.Wu M. Discussion on the construction of cross-border e-commerce ecosystem from the perspective of "Internet +". Business economic research. 2015; 34:75-76.

18.Zhang XX, Zhang X, Zheng X. Research on the Construction of Electronic Commerce Information Ecosystem. Library and Information work. 2010; 54(10):20-24.

19.Cao WJ, Yan MN, Xue CG. Construction of Logistics enterprise-led cross-border E-commerce ecosystem: a multi-case study. Science and technology Management Research. 2019; 39(16):212-222.

20.Xue CG, Li SY, Cao WJ, Cao HW. Construction of payment cross-border e-commerce ecosystem.Monthly finance and accounting magazine. 2019; 19:143-150.

21.Ji SX,Li JY. Study on the evolution and balance of E-commerce ecosystem.Modern Intelligence. 2012; 32(12):71-74.

22.Zhang HN,Xu ZL. Research on the Architecture and evolution of cross-border E-commerce ecosystem.The social sciences. 2020; 02:28-39.

23.Li JB,Zhang Y,Qu F. Research on the construction and development path of cross-border e-commerce ecosystem.Science and technology Management Research. 2019; 39(23): 207-212.

24.Mohammadfam I, Khajevandi AA, Dehghani H, Babamiri M, Farhadian M. Analysis of Factors Affecting Human Reliability in the Mining Process Design Using Fuzzy Delphi and DEMATEL Methods. Sustainability. 2022; 14(13).

25.Si SL, You XY, Liu HC, Zhang P. DEMATEL Technique: A Systematic Review of the State-of-the-Art Literature on Methodologies and Applications. Mathematical Problems in Engineering. 2018.

26.Azzah A, Lazim A, Ahmad TAG, Nur AHA, Mohammad FA. A fusion of decision-making method and neutrosophic linguistic considering multiplicative inverse matrix for coastal erosion problem. Soft Computing. 2019; 24 (13): 9595-9609.

27.Toktaş P, Can GF. A three-stage holistic risk assessment approach proposal based on KEMIRA-M and DEMATEL integration. Knowledge and Information Systems. 2022; 65(4): 1735-1768.

28.Mohammad DE, Ali N, Daria JK, Mehrbakhsh N, Saeed A. Social media addiction: Applying the DEMATEL approach. Telematics and Informatics, 2019, 43(C).

29.Mamta P, Ratnesh L, Prateek P. Application of Fuzzy DEMATEL Approach in Analyzing Mobile App Issues. Programming and Computer Software. 2019; 45(5):268-287.

30.Abid H, Sushil, Mohammad AQ, Sanjay K. Analysis of critical success factors of world-class manufacturing practices: an application of interpretative structural modelling and interpretative ranking process. Production Planning & Control. 2012; 23(10-11): 722-734.

31.Zhu L, Chen JY,Yuan JF. Research on key influencing factors of prefabricated building supply chain resilience based on ISM.Journal of Civil Engineering and Management. 2020; 37(05): 108-114.

32.Han YG, Zhou RD, Geng ZQ, Bai J, Ma B, Fan JZ. A novel data envelopment analysis cross-model integrating interpretative structural model and analytic hierarchy process for energy efficiency evaluation and optimization modeling: Application to ethylene industries. Journal of Cleaner Production. 2020; 246(C).

33.L. AK, Hemalatha J. Modelling e-business influencing factors for supply chain performance of Indian MSMEs: an ISM approach. International Journal of Process Management and Benchmarking. 2022; 12(1).

34.Xie YF, Lv X, Liu R, Mao LY, Liu XX. Research on port ecological suitability evaluation index system and evaluation model. Frontiers of Structural and Civil Engineering. 2015; 9 (1): 65–70.

35.Tavana M, Izadikhah M, Saen RF, Zare R. An integrated data envelopment analysis and life cycle assessment method for performance measurement in green construction management. Environmental science and pollution research international. 2021; 28 (1): 664–682.

36.Yu JF, Shi XB. Research on Innovation Ability Evaluation of Agricultural Machinery Equipment Enterprises based on tomographic Analysis. Scientific Management Research. 2022; 40(06): 100-106.

37.RezaHoseini Ali,Ahmadi Elmira,Saremi Pantea,BagherPour Morteza. Implementation of Building Information Modeling (BIM) Using Hybrid Z-DEMATEL-ISM Approach[J]. Advances in Civil Engineering,2021,2021.

38.Li YH, Yuan YW. Research on low-carbon Transformation Mechanism of Closed-loop Supply Chain of Manufacturing Enterprises under Carbon neutral Objective: Based on DEMATEL-ISM Model. Science and technology management Research. 2022; 42(23): 226-234.

39.Qiu L, Hong JZ. Composition and Development Strategy of Cross-border E-commerce ecosystem in China.Business Economic Research. 2019; 05:126-128.

40.Liu MY. Research on the influencing factors of agricultural product e-commerce ecosystem development in Guangxi.scholarly journal.M.Sc. Thesis, Guangxi University for Nationalities. 2019.”

Thank you again for the suggestions and comments, which have further improved our paper.

Responses to Comments of Reviewer #2

We would like to thank you for your time spent in reviewing our paper and for providing us with valuable comments and suggestions. We also appreciate your positive view on our results and hope that this version has fully addressed your concerns. To make it easy to follow, we first show (in italic) your comment and then present our response (in normal font).

Comments

Comment #1 Please provide the weakness in the related work.

Response/Revision: Thanks for comments! In the literature review section of our paper, we have summarized the relevant literature and identified three weaknesses of previous research based on our analysis of the literature (In lines 117-123). Please see the details bellow.

“Based on the above, previous research on the cross-border e-commerce ecosystem has established a preliminary foundation. However, further research has revealed the following shortcomings: 1. Lack of comprehensive summarization and analysis of the factors influencing the development of the cross-border e-commerce ecosystem; 2. There are still many research gaps in the interactive relationships among the relevant influencing factors; 3. There is a lack of quantitative analysis of the relevant influencing factors. ”

Comment #2 Why and how does this study use many methods?

Response/Revision: We are grateful for your approval of this paper. In Section 3.1 of the article, we expanded the description and analysis of our methods (In lines 156-182). First, we introduced the two methods used in this study and then provided a description of traditional methods such as the PSR theory framework and Data Envelopment Analysis, while highlighting the limitations of these traditional methods in this study. Furthermore, we added a detailed description of the specific research steps taken in applying the two methods used in this study, comparing them to the drawbacks of using only one method. We also provided an explanation of the advantages of using two methods. Please see the details bellow.

“In order to promote and enhance the development of the cross-border e-commerce ecosystem, this study provides a comprehensive summary and analysis of the factors that affect its development, and further analyses the interactive relationship and degree of the influencing factors, identifies key factors, causal relationships between factors, and the hierarchical influence structure. However, traditional research methods have limitations, such as the PSR theory framework [34], data envelopment analysis method [35], analytic hierarchy process [36], and ISM method [32], have limitations. The PSR theory framework method is a qualitative method with poor objectivity. The data envelopment analysis method requires a high sample size. The use of the ISM method alone can only identify the hierarchical relationship between factors. Therefore, this study adopts a combination of the DEMATEL and ISM methods to achieve complementary advantages, clarifying the importance and causality of each factor in the system, and deepening the understanding of the logical relationships and hierarchical structures between factors.

Specifically, this study combines the comprehensive influence matrix in DEMATEL with the unit matrix to obtain the overall influence matrix, and transforms it into the reachable matrix required by ISM through calculation [37]. Compared with using ISM alone, this method not only shows the relationship between influencing factors, but also reflects the strength of interaction between them. The DEMATEL method is micro-oriented while the ISM method is macro-oriented [38]. Integrating the DEMATEL and ISM methods for research can complement advantages, improve computing efficiency, and comprehensively analyze the influencing factors of the cross-border e-commerce ecosystem from the levels, paths, and degrees of influence. This combination method avoids the shortcomings of DEMATEL in expressing the interrelationships and logical relationships between influencing factors and the shortcomings of ISM in accurately analyzing the degree of influence of each influencing factor on the complex system.”

Comment #3 Please provide the profile of the experts in the study. how do you get the data?

Response/Revision: We are grateful for your approval of this paper. First, In the supporting information section, we have included detailed profiles of relevant experts and questionnaire results, In the fourth section of our article, we have added relevant descriptions (In lines 256-263). In order to get the data, our literature sources were obtained from research websites such as Scopus, Web of Science, Google Scholar, and CNKI, with search dates ranging from December 2012 to December 2022. And we have described in detail the process by which the experts were involved in selecting the indicators. In section 5.1, we have further described the assessment of the interaction between the indicators by the members of the expert group (In lines 293-304). Please see the details bellow.

“Initially, literature search was conducted on various academic search engines, such as Scopus, Web of Science, Google Scholar, and CNKI, using a combination of keywords such as "cross-border electronic commerce," "ecosystem," "influence mechanism of e-commerce ecosystem," and their combinations. Relevant literature from the last decade was selected, with the search timeframe ranging from December 2012 to December 2022. Core journal articles and highly cited papers related to the influencing factors of "cross-border e-commerce ecosystem" were then organized, summarized, and analysed. The main references for this study are references [2,11,39, and 40]. A typical scientific, comprehensive, and practical analysis of influencing factors was carried out and the results were refined and combined through expert interviews to form a preliminary indicator system. Subsequently, seven experts were invited to participate in the refinement of the preliminary indicator system. The panel was composed of three academic experts in the field of cross-border e-commerce (all holding the rank of associate professor or higher), two practitioners with over five years of experience in cross-border e-commerce, and two relevant government officials. The panel reviewed the initial theoretical indicators and made adjustments based on the following criteria: (1) if one expert suggested adding a specific indicator, the entire panel would discuss and determine whether to include it; (2) indicators deemed unimportant by two or more experts were deleted; and (3) in cases of disagreement, the panel would collectively discuss whether to modify or delete the indicator. Based on expert discussions and opinions, the influencing factors were divided into four primary factors: internal enterprise, policy support, economic foundation, and external environment, and subdivided into 19 specific influencing factors as shown in Table 1.

5.1 Importance ranking of influencing factors and causal relationship analysis based on the DEMATEL method

This study primarily collected data through survey and interview methods. After conducting research on experts, teachers in the field of cross-border e-commerce, and leaders of management departments related to e-commerce, an expert team of 11 members was invited to participate in the study. In addition to the 3 experts from academic research in cross-border e-commerce (all holding associate professor or higher titles), 2 professionals with over 5 years of experience in cross-border e-commerce related industries, 2 relevant government officials mentioned in Chapter 4, and 4 cross-border e-commerce consumers who have been purchasing goods through cross-border e-commerce for more than 5 years were added to the team. The expert team used interview and Delphi methods to evaluate the interaction of factors, and the scoring system was based on a five-point scale ranging from 0 to 4 (0 indicating no influence, and 4 indicating very strong influence).”

Comment #4 Please provide the Multi-level hierarchical structure model of the influencing the first-order factors of the cross-border e-commerce ecosystem

Response/Revision: Thanks! In section 5.2, we constructed a multi-level hierarchical structure model of first-level impact factors based on the multi-level hierarchical structure model of cross-border e-commerce ecosystem influence factors in Figure 3. This model is shown in Figure 4, and we provided a description of the reasons for constructing a multi-level hierarchical structure model of first-level influencing factors (In lines 396-408). Please see the details bellow.

Fig 4. Multi-level hierarchical structure model of first-level influencing factors of cross-border e-commerce ecosystem.

“Based on the multi-level hierarchical structure model of the influencing factors of cross-border e-commerce ecosystem shown in Fig 3, we construct a multi-level hierarchical structure model of first-level influencing factors. Examining the superficial layer of Fig 3, we can see that the influencing factor indicators mostly belong to the first-level indicators of policy support and economic foundation. Therefore, we assign policy support and economic foundation to the first layer. The second layer of Fig 3 includes influencing factors that mostly belong to the first-level indicator of external environment; hence, we group external environment into the second layer. Similarly, the third and fourth layers of Fig 3 consist mostly of influencing factors belonging to the first-level indicator of internal enterprise, so we assign this indicator to the third layer of Fig 4. Following this logical pattern, we can construct a multi-level hierarchical structure model of first-level influencing factors, as shown in Fig 4.”

Comment #5 Please compare the results of fig. 2 and Fig 3.

Response/Revision: Thanks for comments! In section 5.4, we have added a comparison between Figure 2 and Figure 3, which demonstrates the consistency of the two models in identifying and categorizing the key factors affecting the cross-border e-commerce ecosystem. This comparison also provides evidence for the effectiveness and feasibility of the model proposed in this paper (In lines 423-444). Please see the details bellow.

5.4 Mutual verification of DEMATEL and ISM modeling results

The DEMATEL method identifies key elements and their degree of influence in complex systems through measures such as centrality and causality. On the other hand, the ISM method determines the inherent logical structure and hierarchy of elements in a system. By combining the strengths of both algorithms, this study creates a hierarchical and structured model with greater explanatory power.

Comparing the results of the DEMATEL (Fig 2) and ISM (Fig 3) modeling, we can find correlations and to some extent, consistency between the analysis results of the two methods, thus verifying the accuracy of the model analysis. In the DEMATEL analysis, the factors with the highest influence degree on the development of e-commerce platforms (S1), cross-border e-commerce competitiveness (S17), and cross-border e-commerce transaction scale (S16) belong to the deep-layer factors in the ISM analysis. Conversely, factors with lower influence degrees belong to surface-level factors in the ISM analysis. Moreover, the root-layer influencing factors in the multi-level hierarchical structure model, namely the development of e-commerce platforms (S1), cross-border e-commerce competitiveness (S17), and the competition and cooperation between enterprises (S6), correspond to the set of reasons. By comparing the graphical results, we find that the two models have consistency in identifying the importance and type of influencing factors in the cross-border e-commerce ecosystem, thereby proving the effectiveness and feasibility of the model established in this study.”

Comment #6 Please identify the contribution of this study.

Response/Revision: Thanks for comments! In the sixth section, we elaborated on three key points that highlighted the managerial implications of our findings and the contributions of our study. Additionally, we provided a discussion of future research prospects. This section underscored the practical value of our study (In lines 477-497). Please see the details bellow.

“6.2 Research Contributions

This study has multiple contributions. First, based on the previous work, this study used literature mining and expert interviews to identify factors that affect the development of the cross-border e-commerce ecosystem and determine key factors. Second, this study proposed the DEMATEL-ISM model to study the interaction between factors and provide insights and guidance for cross-border e-commerce management decision-makers. At the same time, this study applied the DEMATEL-ISM model to the cross-border e-commerce ecosystem, providing a reference research paradigm for related studies. In addition, the author hopes that the findings of this study can be used as a reference for the operation and management of the cross-border e-commerce ecosystem. It is hoped that the key indicators affecting the cross-border e-commerce ecosystem identified by this quantitative method can provide important references and decision-making support for its operation and management.

The findings of this study have certain reference significance for cross-border e-commerce ecosystem operation and management. On the basis of a literature review and full investigation, this study identified 19 factors affecting the cross-border e-commerce ecosystem. The importance and correlation of related factors were further analysed. It is hoped that the key indicators affecting the cross-border e-commerce ecosystem identified by this quantitative method can provide an important reference and decision support for its operation management.”

Comment #7 Please update the reference.

Response/Revision: Thanks for comments! We have updated the references. In order to make my paper more communicated with international researchers, we have reorganized the reference section and added English literature closely related to the research, expanding the number of references to 40 (In lines 536-645). Please see the details bellow.

“References

1.Cumming D, Johan S, Khan Z, Meyer M. E-Commerce Policy and International Business. Management international review : MIR : journal of international business. 2022; 63(1): 3-25.

2.Gao TG. Study on the Intention of Foreign Trade Driven by Cross-Border E-Commerce Based on Blockchain Technology. Security and Communication Networks. 2021.

3.Zhang XH. Construction mechanism and implementation path of cross-border e-commerce ecosystem. Contemporary Economic Management. 2021; 43(07): 55-60.

4.Sun LB, Lyu GD, Yu YG, Teo CP. Cross-Border E-commerce Data Set: Choosing the Right Fulfillment Option. Manufacturing & Service Operations Management. 2020; 23(5): 1297-1313.

5.Deng XG, Ouyang YX. Cross-Border Supply Chain System Constructed by Complex Computer Blockchain for International Cooperation. Computational intelligence and neuroscience. 2022.

6.Xue CG, Zhou ML,Cao WJ. Research on dynamic mechanism of cross-border e-commerce ecosystem based on system dynamics. Industrial engineering. 2020; 23(04): 84-92.

7.Zhu JS, Lan WD, Zhang XC. Geographic proximity, supply chain and organizational glocalized survival: China's e-commerce investments in Indonesia. PloS one. 2021; 16(9).

8.Miao M, Krishna J. Mobile payments in Japan, South Korea and China: Cross-border convergence or divergence of business models?. Telecommunications Policy. 2016; 40(2-3):182-196

9.Tikhomirova A, Huang J, Chuanmin S, Khayyam M, Ali H, Khramchenko DS. How Culture and Trustworthiness Interact in Different E-Commerce Contexts: A Comparative Analysis of Consumers' Intention to Purchase on Platforms of Different Origins. Frontiers in Psychology, 2021,12.

10.Du J,Yu ZY. Building a Cross-Border E-Commerce Ecosystem Model Based on Block Chain + Internet of Things. Security and Communication Networks. 2021.

11.He J, Li JJ, Ge L. Model and Simulation of Symbiotic Evolutionary Dynamics of a Marine Cross-Border E-Commerce Trade Ecosystem. JOURNAL OF COASTAL RESEARCH. 2020; 108: 95-98.

12.Wulfert T,Woroch R, Strobel G, Seufert S, Möller F. Developing design principles to standardize e-commerce ecosystems: A systematic literature review and multi-case study of boundary resources. Electronic markets. 2022; 32(4): 1813-1842.

13.Rong K, Zhou D, Shi XW, Huang W. Social Information Disclosure of Friends in Common in an E‐commerce Platform Ecosystem: An Online Experiment. Production and Operations Management. 2021; 31(3): 984-1005.

14.Zhang XH. Jingdong: Build a cross-border e-commerce ecosystem. Enterprise management. 2016; (11):102-104.

15.Peng C, Jing X,Tie J, Tian Y, Kong JY, Xue K, Zhou Y. Research on Value Co-Creation New Business Model of Import Cross-Border E-Commerce Platform Ecosystem. Security and Communication Networks. 2022.

16.You J, Peng LH.Study on ecological characteristics of e-commerce ecosystem. Enterprise economic. 2017; 36(08):115-122.

17.Wu M. Discussion on the construction of cross-border e-commerce ecosystem from the perspective of "Internet +". Business economic research. 2015; 34:75-76.

18.Zhang XX, Zhang X, Zheng X. Research on the Construction of Electronic Commerce Information Ecosystem. Library and Information work. 2010; 54(10):20-24.

19.Cao WJ, Yan MN, Xue CG. Construction of Logistics enterprise-led cross-border E-commerce ecosystem: a multi-case study. Science and technology Management Research. 2019; 39(16):212-222.

20.Xue CG, Li SY, Cao WJ, Cao HW. Construction of payment cross-border e-commerce ecosystem.Monthly finance and accounting magazine. 2019; 19:143-150.

21.Ji SX,Li JY. Study on the evolution and balance of E-commerce ecosystem.Modern Intelligence. 2012; 32(12):71-74.

22.Zhang HN,Xu ZL. Research on the Architecture and evolution of cross-border E-commerce ecosystem.The social sciences. 2020; 02:28-39.

23.Li JB,Zhang Y,Qu F. Research on the construction and development path of cross-border e-commerce ecosystem.Science and technology Management Research. 2019; 39(23): 207-212.

24.Mohammadfam I, Khajevandi AA, Dehghani H, Babamiri M, Farhadian M. Analysis of Factors Affecting Human Reliability in the Mining Process Design Using Fuzzy Delphi and DEMATEL Methods. Sustainability. 2022; 14(13).

25.Si SL, You XY, Liu HC, Zhang P. DEMATEL Technique: A Systematic Review of the State-of-the-Art Literature on Methodologies and Applications. Mathematical Problems in Engineering. 2018.

26.Azzah A, Lazim A, Ahmad TAG, Nur AHA, Mohammad FA. A fusion of decision-making method and neutrosophic linguistic considering multiplicative inverse matrix for coastal erosion problem. Soft Computing. 2019; 24 (13): 9595-9609.

27.Toktaş P, Can GF. A three-stage holistic risk assessment approach proposal based on KEMIRA-M and DEMATEL integration. Knowledge and Information Systems. 2022; 65(4): 1735-1768.

28.Mohammad DE, Ali N, Daria JK, Mehrbakhsh N, Saeed A. Social media addiction: Applying the DEMATEL approach. Telematics and Informatics, 2019, 43(C).

29.Mamta P, Ratnesh L, Prateek P. Application of Fuzzy DEMATEL Approach in Analyzing Mobile App Issues. Programming and Computer Software. 2019; 45(5):268-287.

30.Abid H, Sushil, Mohammad AQ, Sanjay K. Analysis of critical success factors of world-class manufacturing practices: an application of interpretative structural modelling and interpretative ranking process. Production Planning & Control. 2012; 23(10-11): 722-734.

31.Zhu L, Chen JY,Yuan JF. Research on key influencing factors of prefabricated building supply chain resilience based on ISM.Journal of Civil Engineering and Management. 2020; 37(05): 108-114.

32.Han YG, Zhou RD, Geng ZQ, Bai J, Ma B, Fan JZ. A novel data envelopment analysis cross-model integrating interpretative structural model and analytic hierarchy process for energy efficiency evaluation and optimization modeling: Application to ethylene industries. Journal of Cleaner Production. 2020; 246(C).

33.L. AK, Hemalatha J. Modelling e-business influencing factors for supply chain performance of Indian MSMEs: an ISM approach. International Journal of Process Management and Benchmarking. 2022; 12(1).

34.Xie YF, Lv X, Liu R, Mao LY, Liu XX. Research on port ecological suitability evaluation index system and evaluation model. Frontiers of Structural and Civil Engineering. 2015; 9 (1): 65–70.

35.Tavana M, Izadikhah M, Saen RF, Zare R. An integrated data envelopment analysis and life cycle assessment method for performance measurement in green construction management. Environmental science and pollution research international. 2021; 28 (1): 664–682.

36.Yu JF, Shi XB. Research on Innovation Ability Evaluation of Agricultural Machinery Equipment Enterprises based on tomographic Analysis. Scientific Management Research. 2022; 40(06): 100-106.

37.RezaHoseini Ali,Ahmadi Elmira,Saremi Pantea,BagherPour Morteza. Implementation of Building Information Modeling (BIM) Using Hybrid Z-DEMATEL-ISM Approach[J]. Advances in Civil Engineering,2021,2021.

38.Li YH, Yuan YW. Research on low-carbon Transformation Mechanism of Closed-loop Supply Chain of Manufacturing Enterprises under Carbon neutral Objective: Based on DEMATEL-ISM Model. Science and technology management Research. 2022; 42(23): 226-234.

39.Qiu L, Hong JZ. Composition and Development Strategy of Cross-border E-commerce ecosystem in China.Business Economic Research. 2019; 05:126-128.

40.Liu MY. Research on the influencing factors of agricultural product e-commerce ecosystem development in Guangxi.scholarly journal.M.Sc. Thesis, Guangxi University for Nationalities. 2019.”

Thank you again for the suggestions and comments, which have further improved our paper.

---

## [Decision Letter · Decision Letter 1]

5 Jun 2023

Analysis of the key influencing factors of China's cross-border e-commerce ecosystem based on the DEMATEL-ISM method

PONE-D-23-03573R1

Dear Dr. Xi,

We’re pleased to inform you that your manuscript has been judged scientifically suitable for publication and will be formally accepted for publication once it meets all outstanding technical requirements.

Kind regards,

Tinggui Chen

Academic Editor

PLOS ONE

Additional Editor Comments (optional):

Reviewers' comments:

Reviewer's Responses to Questions

**Comments to the Author**

1. If the authors have adequately addressed your comments raised in a previous round of review and you feel that this manuscript is now acceptable for publication, you may indicate that here to bypass the “Comments to the Author” section, enter your conflict of interest statement in the “Confidential to Editor” section, and submit your "Accept" recommendation.

Reviewer #1: All comments have been addressed

Reviewer #2: All comments have been addressed

2. Is the manuscript technically sound, and do the data support the conclusions?

Reviewer #1: Yes

Reviewer #2: Yes

3. Has the statistical analysis been performed appropriately and rigorously? 

Reviewer #1: Yes

Reviewer #2: Yes

4. Have the authors made all data underlying the findings in their manuscript fully available?

Reviewer #1: Yes

Reviewer #2: Yes

5. Is the manuscript presented in an intelligible fashion and written in standard English?

Reviewer #1: Yes

Reviewer #2: Yes

6. Review Comments to the Author

Reviewer #1: The authors have completed the revisions according to the review suggestions, and the quality of the paper has achieved some improvement.

Reviewer #2: The revised results show that three factors, i.e., the e-commerce platform development level, cross-border ecommerce competitiveness, and the cross-border e-commerce transaction scale, have a greater degree of influence on the other influencing factors. These revised manuscript has adressed my concern.

7. PLOS authors have the option to publish the peer review history of their article (what does this mean?). If published, this will include your full peer review and any attached files.

Reviewer #1: No

Reviewer #2: No

---

## [Editor Report · Acceptance letter]

3 Aug 2023

PONE-D-23-03573R1 

Analysis of the key influencing factors of China's cross-border e-commerce ecosystem based on the DEMATEL-ISM method 

Dear Dr. Xi:

I'm pleased to inform you that your manuscript has been deemed suitable for publication in PLOS ONE. Congratulations! Your manuscript is now with our production department. 

Kind regards, 

on behalf of

Dr. Tinggui Chen 

Academic Editor

PLOS ONE